# Detection of multiple mycetoma pathogens using fungal metabarcoding analysis of soil DNA in an endemic area of Sudan

Hiroki Hashizume[1,2☺], Suguru Taga[2☺], Masayuki K. Sakata[3], Mahmoud Hussein Mohamed Taha[4], Emmanuel Edwar Siddig[4], Toshifumi Minamoto[3], Ahmed Hassan Fahal[4], Satoshi Kaneko[1,2¤]*

**1** School of Tropical Medicine and Global Health, Nagasaki University, Nagasaki, Japan, **2** Department of Ecoepidemiology, Institute of Tropical Medicine (NEKKEN), Nagasaki University, Nagasaki, Japan, **3** Graduate School of Human Development and Environment, Kobe University, Kobe, Japan, **4** Mycetoma Research Center, University of Khartoum, Khartoum, Sudan

☺ These authors contributed equally to this work.
¤ Current address: Department of Ecoepidemiology, Institute of Tropical Medicine (NEKKEN), Nagasaki University, Nagasaki, Japan
* skaneko@nagasaki-u.ac.jp

**Data Availability Statement:** Raw reads generated during the current study are available in the DDBJ

## Abstract

Mycetoma is a tropical disease caused by several fungi and bacteria present in the soil. Fungal mycetoma and eumycetoma are especially challenging to treat; therefore, prevention, early diagnosis, and early treatment are important, but it is also necessary to understand the geographic distribution of these pathogenic fungi. In this study, we used DNA metabarcoding methodology to identify fungal species from soil samples. Soil sampling was implemented at seven villages in an endemic area of Sennar State in Sudan in 2019, and ten sampling sites were selected in each village according to land-use conditions. In total, 70 soil samples were collected from ground surfaces, and DNA in the soil was extracted with a combined method of alkaline DNA extraction and a commercial soil DNA extraction kit. The region for universal primers was selected to be the ribosomal internal transcribed spacer one region for metabarcoding. After the second PCR for DNA library preparation, the amplicon-based DNA analysis was performed using next-generation sequencing with two sets of universal primers. A total of twelve mycetoma-causative fungal species were identified, including the prime agent, *Madurella mycetomatis*, and additional pathogens, *Falciformispora senegalensis* and *Falciformispora tompkinsii*, in 53 soil samples. This study demonstrated that soil DNA metabarcoding can elucidate the presence of multiple mycetoma-causative fungi, which may contribute to accurate diagnosis for patient treatment and geographical mapping.

## Author summary

Mycetoma, a chronic subcutaneous and cutaneous disease, designated as a "neglected tropical disease," is prevalent in dry and hot climates. Fungal mycetoma is caused by more

Sequence Read Archive (DRA) under the accession numbers: DRA012568 and DRA013345.

**Funding:** This study was supported by JSPS KAKENHI with grant ID 18K19684 and 21K19656 (Japan Society for the Promotion of Science, https://www.jsps.go.jp) to S.K; and AMED under Grant Number JP21jm0510005 (Japan Agency for Medical Research and Development, https://www.amed.go.jp) to S.K. The funders had no role in study design, data collections and analysis, decision to publish, or preparation of the manuscript.

**Competing interests:** The authors have declared that no competing interests exist.

than 50 species of soil-dwelling pathogenic fungi, and its diagnosis and treatment can be challenging. The prevention of infection and early diagnosis and treatment are essential, and for this purpose, environmental assessment to understand the fungal habitat is necessary. In this study, we performed DNA metabarcoding analysis using next-generation sequencing (NGS) for mycetoma pathogens from environmental soil samples in Sudan. The results suggest that multiple causative agents of fungal mycetoma are widespread regardless of the environment and can be a source of infection anywhere in an endemic area. Based on the results of this study, we expect that the investigation of fungi in soil using NGS technology may help identify infection routes and create risk maps for the prevention of mycetoma.

## Introduction

Mycetoma is a chronic granulomatous and disabling inflammatory disease caused by specific groups of bacteria (actinomycetoma) or fungi (eumycetoma). It typically affects people living in poor, remote communities in tropical and subtropical regions within the so-called mycetoma belt, located between latitude 15˚S and 30˚N [1]. Most of the causative microorganisms inhabit the soil and invade the human body through minor unnoticed wounds on the skin, mainly in the foot and hand, and multiply to form multiple painless subcutaneous mass lesions that discharge seropurulent grains [1–4]. As the lesions progress, the microorganisms invade more deeply into tissues and bones, which can lead to amputation of the affected limb and is sometimes fatal [5]. The detailed epidemiological characteristics, such as the route of transmission, incubation period, prevalence, and incidence, have not been elucidated due to its chronic slow progression, with patients only visiting the hospital after being infected for years [6,7].

Between actinomycetoma and eumycetoma, the latter causes serious public health problems, because there is no effective short-term medicine compared to the former, which is treated by existing antibiotics. Therefore, prevention and screening programs are needed for early detection and treatment before the lesions reach vital parts of the body. It is thus necessary to understand the geographical risk distribution of the infective fungi. Because the causative fungal pathogens exist in the soil in endemic areas, it is possible to determine the distribution of infection by analyzing collected soil samples from areas in or near-endemic areas to confirm their presence. Before now, only soil sampling surveys have been conducted explicitly targeting only *Madurella mycetomatis*, the leading causative agent in Sudan [4]. However, to date, over 50 species of mycetoma-causative fungi have been reported around the world [8], and some patients are simultaneously infected with multiple fungi [9–13]. To understand the distributions of the causative fungi, it is necessary to comprehensively capture all fungi in an endemic area's soil.

To comprehensively capture and analyze large quantities of DNA information simultaneously, metabarcoding using next-generation sequencing (NGS) is used as a high throughput approach in several scientific fields. This technique has been applied for soil microbiome research to detect environmental DNA (eDNA) from soil samples in the fields of soil microbial ecology, environmental science, and botany [14–16]. By applying this metabarcoding technology, it is possible to capture all the causative fungi in the soil for environmental surveillance of mycetoma, which will be critical for eumycetoma prevention measurements and control programs.

In this study, we performed a metabarcoding technology-based soil sampling survey in an endemic area to establish the geographical risk distribution of eumycetoma.

## Methods

### Study area and soil sampling

We chose ten villages in the state of Sennar, Sudan, located about 250 km south-east of Khartoum, where a mycetoma clinic is operated by the Mycetoma Research Centre (MRC), University of Khartoum, to manage mycetoma patients (Fig 1). Ten sampling sites where people gather or were near water sources were selected per village; we planned 100 soil sample collections from 10 villages. Each sampling site was categorized according to the land usage at the site as follows: 1) inside a cow fence, 2) dryland, 3) farmland, 4) riverside farm, and 5) road. The sampling survey was conducted for two days using a mobile data collection system, Open Data Kit (available online: https://opendatakit.org/) [17]. The soil sampling survey was conducted from October 16 to 17, 2019. The average temperature of Senner in October of 1991–2020 was 29.64˚C (range: 22.28˚C–37.05˚C), and the average precipitation was 24.11 mm (sourced from the World Bank Climate Change Knowledge Portal) [18]. Soil samples were

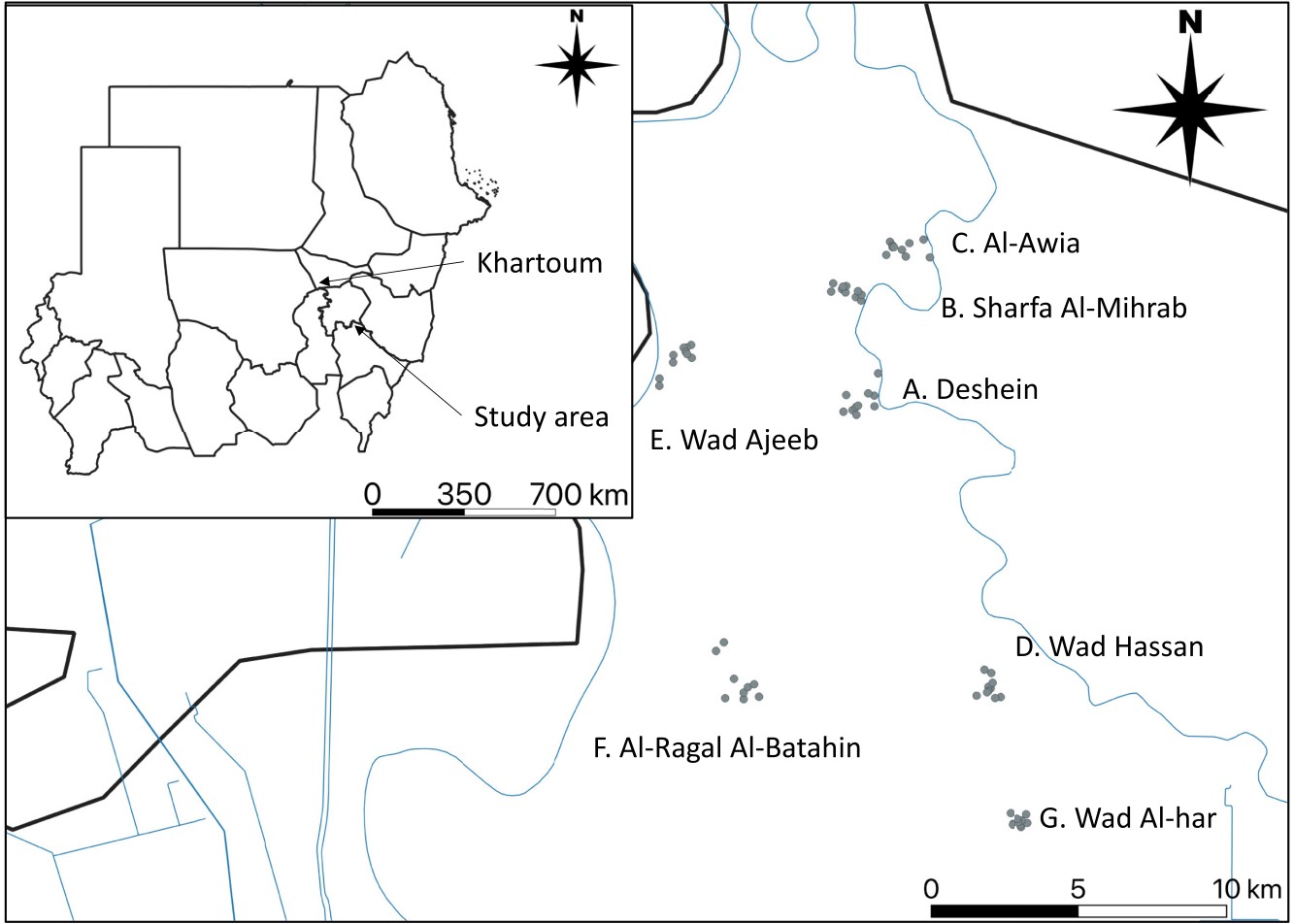

**Fig 1. The geographical location of areas where soil samples were collected in this study.** Each point on the map presents a sample collection site. Country and state maps were obtained from the GADM database under a CC BY license (https://geodata.ucdavis.edu/gadm/gadm4.0/shp/gadm40_SDN_shp.zip). Waterways (blue line) data were downloaded from the OpenStreetMap project (OpenStreetMap contributors) under a CC BY-SA 2.0 license (www.openstreetmap.org) through the platform Geofabrik (https://download.geofabrik.de/africa/sudan-latest-free.shp.zip). The map was created using the QGIS Geographic Information System, Open Source Geospatial Foundation Project, under a CC BY-SA 3.0 license (http://qgis.osgeo.org).

collected with 50 ml Falcon tubes from surfaces that none of the team members had stepped on using disposable plastic shovels [19]. The gloves and disposable shovels were changed at every location, and shoe covers were worn at every village to avoid contamination. The Falcon tubes were packed in a small plastic bag and stored in a styrofoam box with dry ice to minimize DNA degradation. Samples from the first day were kept in an empty −30˚C freezer provided by the village head. After completing sampling on the second day, soil samples in the tubes were packed in styrofoam boxes with additional dry ice, transported to Khartoum by land, then stored in a −30˚C freezer immediately upon arrival at MRC (see Supplemental method 1 in S1 Protocol for more details).

## DNA extraction

The DNA extraction protocol was broadly followed as previously reported to increase the amount of DNA extracted from the soil samples [19,20]. We used the integrated method of alkaline DNA extraction with ethanol precipitation and a commercial DNA extraction kit for soil samples (PowerSoil DNA Isolation Kit, Qiagen, Germany). For each sample, 9 g of soil was used for DNA extraction. During this process for each village, one negative control was obtained using 9 g of distilled water. The final DNA solutions were dissolved into the elusion buffer of the kit (see Supplemental method 2 in S1 Protocol for more details).

## Universal primers

To choose universal primers for metabarcoding on the MiSeq platform (Illumina, USA), we cataloged mycetoma-causing fungi species from previous studies, which resulted in 29 genera with 55 species (including three at genus-level classification) (S1 Table). The internal transcribed spacer (ITS) regions and 18S ribosomal DNA are generally used for the identification of fungal species [21]. The accession of each causative fungus in the GenBank database by the National Center for Biotechnology Information (NCBI) was checked, and subsequently, the ITS1 and ITS2 regions were chosen as universal primer targets (S1 Table). Then, the sequences of 52 species available in GenBank were downloaded and subjected to multiple alignments using MUSCLE in the Unipro UGENE program (version 40.0) [22,23]. The set of aligned sequences was inspected visually with primers reported previously [24–26]. The universal primers that covered as many species as possible in the data set were adopted as our MiSeq primers. Primers sets named ITS1/ITS2 and ITS3_-KYO1/ITS4_KYO1 shared sequences well with the target fungal genes [24,26]. Therefore, primers were designed for MiSeq in combination with ITS1/ITS2 and ITS3_KYO1/ITS4_KYO1, and six random hexamers (N) and adapter sequences (Fig 2 and Table 1).

## PCR amplification

Targeting ribosomal ITS1 and ITS2, we employed two-step PCR library preparations. The first was to amplify specific regions in eumycetoma-causing species, and the second was for the attachment of distinguishable tags for metabarcoding, as reported previously [27,28]. PCR reagents and PCR products were prepared in separate rooms to avoid unanticipated DNA contamination. Three replicates were amplified for the first-round PCR (1st PCR) for each sample using the forward and reverse primers targeting the fungal ITS1 and ITS2 regions (Table 1). A total of 25 μl PCR reaction volume contained 0.5 U of KOD-Plus-Neo (Toyobo, Japan), 2.5 μl of 10× buffer for KOD-Plus-Neo (Toyobo), 2.5 μl of 2 mM dNTPs solution, 1.5 μl of 25 mM MgSO4 solution, 0.75 μl of 10 μM each primer, and 2 μl DNA extract. For the PCR targeting ribosomal ITS2, ten-fold dilutions of 17 colored DNA extracts were also used to reduce the influence of PCR inhibition (S2 Table). The PCR was carried out with 40 cycles at 94˚C for 20 s, 65˚C for 30 s (57˚C for PCR targeting ribosomal ITS2), and 68˚C for 30 s, then completed

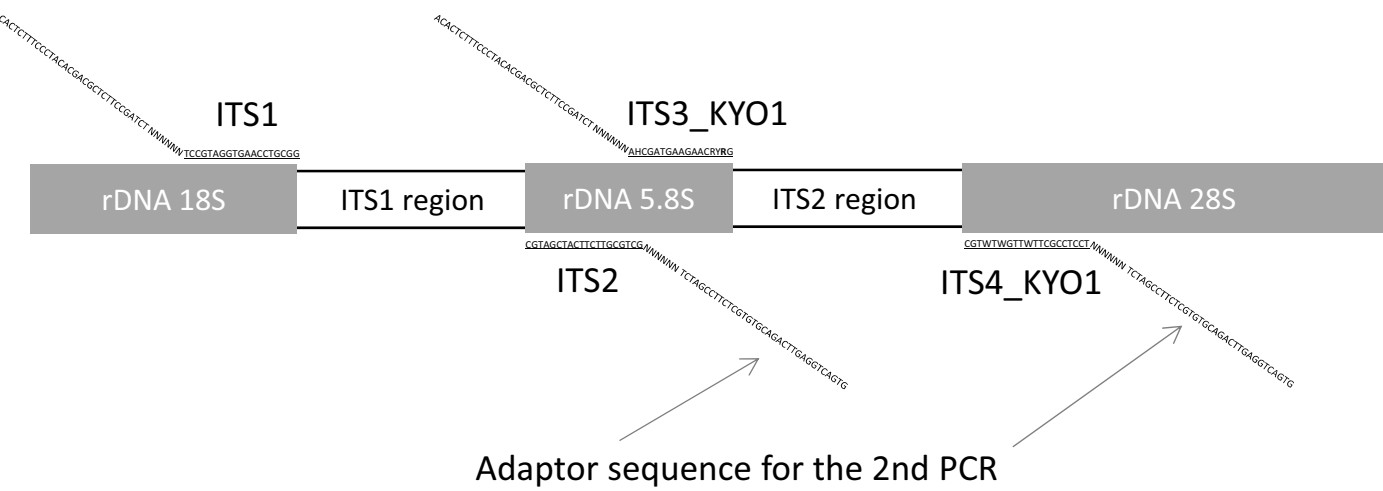

**Fig 2. Map of ribosomal DNA genes with universal primers.** The internal transcribed spacer 1 (ITS1) and 2 (ITS2) regions were targeted to identify causative species. The primer sets ITS1/ITS2 and ITS3_KYO1/ITS4_KYO1 were used as universal primers for the MiSeq system.

with a final 68°C for 5 min. Then, DNA purification was performed for the 1st PCR products with SPRIselect Reagent Kit (Beckman Coulter, USA) and quantified using a Qubit fluorometer 3.0 (Thermo Fisher Scientific, USA). Preliminary checks using gel electrophoresis were performed to confirm PCR products. The dilution factor was calculated for each sample to 0.1 ng/µl, and the average dilution rate was applied to the negative controls. Subsequently, all DNA samples from the soils were diluted for the second-round PCR (2nd PCR).

Next, the 2nd PCR, which added a unique 8-bp index and MiSeq adaptor sequences at each end of the amplicons, was performed for Illumina MiSeq. The PCR was carried out in a 12 µl reaction volume using 6 µl of 2× KAPA HiFi HotStart ReadyMix (KAPA Biosystems, USA), 2 µl of forward and reverse primers with index and adaptor sequences (1.8 µM), 1 µl of combined 1st PCR DNA templates, and 1 µl of ultrapure water under the following thermal cycler profile: 95°C for 3 min, followed by 12 cycles of 98°C for 20 s and 72°C for 30 s, finally 72°C for 5 min. All the 1st PCR products were combined into one and then applied to the MiSeq Reagent Kit v3 for 2 × 300 bp (600 cycles) (Illumina) with PhiX Control v3 (Illumina). Detailed protocols are described in the supplement (see Supplemental method 3 in S1 Protocol).

### Bioinformatics

According to previously reported methods, the raw MiSeq data were pretreated and analyzed using USEARCH v10.0.240 [19,29]. Paired-end reads were combined; meanwhile, reads with

**Table 1. Primer sequences for the MiSeq analysis.**

| Primers | Sequence | Original primer names |
|---|---|---|
| ITS1_U | 5'-ACACTCTTTCCCTACACGACGCTCTTCCGATCT NNNNNN <u>TCCGTAGGTGAACCTGCGG</u>-3' | ITS1[a] |
| ITS2_U | 5'-GTGACTGGAGTTCAGACGTGTGCTCTTCCGATCT NNNNNN <u>GCTGCGTTCTTCATCGATGC</u>-3' | ITS2[a] |
| ITS3_KYO1_U | 5'- ACACTCTTTCCCTACACGACGCTCTTCCGATCT NNNNNN <u>AHCGATGAAGAACRY**R**G</u>- 3' | ITS3_KYO1[b] |
| ITS4_KYO1_U | 5'- GTGACTGGAGTTCAGACGTGTGCTCTTCCGATCT NNNNNN <u>TCCTCCGCTTWTTGWTWTGC</u>- 3' | ITS4_KYO1[b] |

Underlined sequences are traditionally used as universal fungal primers. The bolded base was slightly modified from the original.

[a] reference [24]

[b] reference [26]

low quality, short length, and many differences (>5 positions) in the merged region were discharged. Next, primer sequences, low-quality reads, and short reads were removed. Dereplication was performed to the set of reads; then denoising was conducted to generate amplicon sequence variants (ASVs). Chimeric and minor (<10 reads) ASVs were removed.

Then the data were analyzed systematically and robustly, and species-level identification by ASVs was conducted through BLAST+ [30] searches with a 97% identity threshold and 90% query cover of the entire query sequence [27,31,32]. Since a small number of ASVs occurred in negative control samples, these ASV sequences were first removed from the sequence data of all samples. We downloaded and used the UNITE database v8.2 (2020-02-04) (https://unite.ut.ee/) [33,34]. After a homology search with BLASTN, the species names of the eumycetoma-causative fungi were extracted from the results. To ensure identification at the level of a single species, we checked the species names manually, whether best (i.e., single and top) or not, with reference to the value of identity and E-values (see Supplemental method 4 in S1 Protocol for more details).

## Statistical analysis

One-way analysis of variance (ANOVA) followed by the Turkey-Kramer test was conducted to test whether there were differences in the number of causative species by land use of the sampling sites, using R (version 4.1.2, 64-bit).

## Results

### Soil sampling and DNA extraction

Finally, 70 soil samples were retrieved from seven villages (Al Awia, Al Ragal Al Batahin, Deshein, Sharfa Al Mihrab, Wad Ajeeb, Wad Al-Har, Wad Hassan) (Fig 3). We could not approach three villages due to flooding. Although the weather was sunny during the sampling, one village (Wad Al-Har) was muddy, except for the center of the village. Few riverside farm samples were collected from the targeted location due to secured blockades and fences; therefore, samples were collected from the closest location.

A total of 64 DNA samples with six negative controls were extracted, and 17 of the samples were colored with soil components. During the laboratory work process, seven samples, including one negative control, were lost due to a malfunction of the centrifuge separator (Fig 3).

### PCR amplification and MiSeq sequencing

After the 1st PCR targeting ribosomal ITS1 to amplify fungal DNA, preliminary checks found target size bands from 37 samples, but no band was found in the negative controls. Hence, for the MiSeq for ribosomal ITS2, we did only an initial check for some samples.

As an outcome of MiSeq for ribosomal ITS1, 3,158,851 pair-end sequences (reads) were obtained in total from the soil and negative control samples. The averages (±SD) of soil samples and negative controls (blanks of DNA extraction and the 1st PCR) were 24,631 (± 22,865) reads (lowest number 19 reads; highest number 161,247 reads) and 306 (± 346) reads, respectively. After processing with USEARCH, 7,357 ASVs were found in all DNA samples. The averages (±SD) of soil samples and negative controls/PCR blanks of the 1st PCR were 114 (± 86.0) reads and 6.1 (± 5.7) reads, respectively. For ribosomal ITS2, a total of 21,177,128 pair-end reads, an average of 130,569 (± 126,731) reads (lowest number 14 reads; highest number 679,229 reads) for soil samples and 1,248 (± 2,208) for negative controls, and a total of 10,780 ASVs, an average of 133 (± 107) reads for soil samples, and 5.8 (± 9.8) reads for negative controls, were obtained.

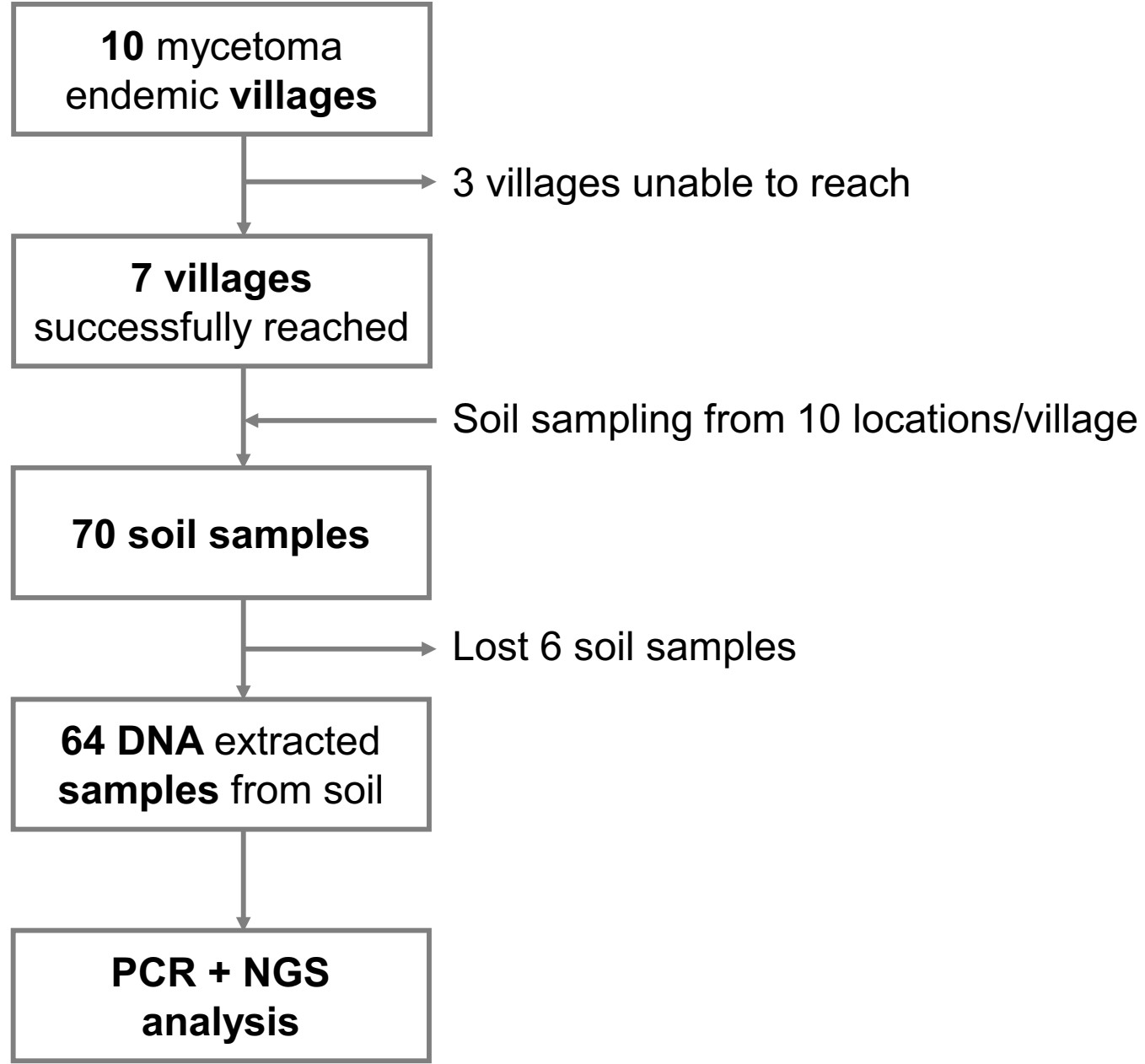

**Fig 3. Schematic diagram of soil sampling prior to NGS analysis.**

### Eumycetoma-causative fungi detected in the samples

In the BLAST results based on the UNITE database, nine genera and twelve species of causative fungi, including the primary pathogen in Sudan, *M. mycetomatis*, and *Falciformispora senegalensis* and *Falciformispora tompkinsii*, were retrieved (Table 2). The number of detected species was counted based on land use; there was a significant difference only between road (3.3 ± 1.5) and dryland (1.9 ± 1.0) (p = 0.045) (S1 Fig).

**Table 2. Causative fungal species hit with 97% identity threshold and top hit species.**

| Village | Land use | Aspergillus terreus | Amesia atrobrunnea | Curvularia lunata | Exserohilum rostratum | Falciformispora senegalensis | Falciformispora tompkinsii | Fusarium solani | Madurella fahalii | Madurella mycetomatis | Madurella tropicana | Medicopsis romeroi | Phaeoacremonium parasiticum |
|---|---|---|---|---|---|---|---|---|---|---|---|---|---|
| A. Deshein | Farm 1 | - | - | c | b | - | - | - | - | - | - | - | - |
| | Farm 2 | - | - | c | a | - | - | - | - | b | - | - | - |
| | Road | - | - | b | b | - | - | b | - | b | - | - | - |
| | Cattle 1 | - | - | a | - | - | - | - | - | - | - | - | - |
| | Cattle 2 | - | b | c | - | - | - | b | - | - | - | - | - |
| | Dryland | - | - | c | - | - | - | - | - | b | - | - | - |
| | Riverfarm | - | - | a | - | - | - | b | - | - | - | - | - |
| B. Sharfa Al-Mihrab | Farm 1 | a | - | - | - | - | - | - | - | - | - | - | - |
| | Road | - | - | a | - | - | - | - | b | b | b | - | - |
| | Dryland 1 | a | - | c | b | - | - | - | b | - | - | - | - |
| | Riverfarm 1 | - | - | a | - | a | - | - | b | - | - | - | - |
| | Riverfarm 2 | - | - | a | - | a | - | - | - | - | - | - | - |
| | Dryland 2 | - | - | - | b | - | - | b | - | - | - | - | - |
| | Farm 2 | - | - | c | - | - | - | b | - | - | - | - | - |
| C. Al-Awia | Farm 1 | - | - | c | - | - | - | - | - | - | - | - | - |
| | Farm 2 | - | - | c | - | - | - | - | - | - | - | - | - |
| | Road 1 | - | - | c | b | - | - | b | - | b | b | - | - |
| | Road 2 | - | b | c | b | - | - | b | - | b | b | - | - |
| | Cattle | - | - | a | - | - | - | - | - | - | b | - | - |
| | Dryland 1 | - | - | a | - | - | - | - | - | - | - | - | - |
| | Riverfarm 1 | - | - | c | - | - | a | b | - | - | - | - | - |
| | Riverfarm 2 | - | - | c | - | - | - | - | - | - | - | - | - |
| | Dryland 2 | - | - | c | - | - | - | - | - | - | - | - | - |
| D. Wad Hassan | Farm 1 | a | - | c | c | - | - | - | - | - | - | - | - |
| | Riverfarm 1 | a | - | c | - | - | - | b | - | - | - | - | - |
| | Road 1 | a | - | c | - | - | - | - | b | - | - | - | b |
| | Road 2 | - | - | b | - | - | - | b | - | b | - | - | - |
| | Farm 2 | - | b | c | - | - | - | b | - | b | - | - | - |
| | Dryland 1 | a | - | c | b | - | - | - | - | - | - | b | - |
| | Dryland 2 | a | - | c | - | - | - | - | - | - | - | - | - |
| | Riverfarm 2 | - | - | c | - | - | - | - | b | - | - | - | - |

*(Continued)*

**Table 2.** (Continued)

| Village | Land use | Aspergillus terreus | Amesia atrobrunnea | Curvularia lunata | Exserohilum rostratum | Falciformispora senegalensis | Falciformispora tompkinsii | Fusarium solani | Madurella fahalii | Madurella mycetomatis | Madurella tropicana | Medicopsis romeroi | Phaeoacremonium parasiticum |
|---|---|---|---|---|---|---|---|---|---|---|---|---|---|
| E. Wad Ajeeb | Farm 1 | a | - | c | - | - | - | - | - | b | - | - | - |
| | Farm 2 | - | - | c | b | - | - | - | - | b | - | - | - |
| | Road 1 | - | - | - | b | - | - | - | b | - | - | - | - |
| | Road 2 | - | - | b | - | - | - | - | - | - | - | - | - |
| | Dryland 1 | - | - | a | - | - | - | - | b | b | - | - | - |
| | Dryland 1 | a | - | - | - | - | - | - | - | - | - | - | - |
| | Riverfarm 1 | - | - | c | - | - | - | b | b | - | - | - | - |
| | Riverfarm 2 | - | - | c | - | - | - | - | - | - | - | c | - |
| F. Al-Ragal Al-Batahin | Dryland | - | - | c | - | - | - | - | - | - | - | - | - |
| | Farm 1 | a | - | c | - | - | - | - | - | - | - | - | - |
| | Farm 2 | a | - | c | b | - | - | - | - | - | - | - | - |
| | Cattle | - | - | a | - | - | - | - | - | - | - | - | - |
| | Road | - | - | b | b | - | - | b | - | b | - | b | - |
| | Farm 1 | - | - | c | - | - | - | b | b | - | - | - | - |
| | Farm 2 | - | - | c | - | - | - | - | - | - | - | - | - |
| G. Wad Al-har | Road 1 | - | - | c | - | - | - | - | b | b | - | - | - |
| | Farm 1 | - | - | c | - | - | - | b | - | b | - | - | - |
| | Farm 2 | - | - | c | - | - | - | - | - | - | - | - | - |
| | Road 2 | - | - | c | - | - | - | - | - | - | - | - | - |
| | Road 3 | - | - | a | - | - | c | - | - | - | - | - | - |
| | Farm 2 | - | - | a | - | - | - | - | - | b | - | - | - |
| | Farm 3 | - | b | c | - | - | - | b | - | - | - | - | - |

Letters indicate the following: a, detected species the MiSeq targeting the ribosomal ITS1 region; b, ribosomal ITS2; c, species hit in both analyses. Only locations where the target fungi were detected are shown.

## Discussion

In this study, we detected several eumycetoma-causing fungi simultaneously from soils of different land uses in an endemic area using a metabarcoding technology-based soil sampling survey method. Our analysis identified twelve species of eumycetoma pathogens from soil DNA samples, including the principal causative agent, *M. mycetomatis*. Only two previous studies detected the DNA of *M. mycetomatis* and other causative species from the soil or *Acacia* thorns and other environmental agents [4,16]. However, the simultaneous detection of multiple pathogenic fungi is scientifically warranted because 1) over 50 species of mycetoma-causative fungi are reported worldwide [8], 2) mycetoma is caused by various fungi classes [5,35], 3) multiple fungi are found even in a single lesion of a patient [36], and 4) it has been reported that fungi collected from patient lesions and cultured are present in the soil [6,7,35].

As previous reports have mentioned, mycetoma is distributed widely within arid areas, called the mycetoma belt [1]. In this study, the field sites were selected from the state of Sennar, which is in a dry climate. Soil samples from several different land-use conditions in the endemic region of eumycetoma were also collected and analyzed. As a result, surprisingly, the DNA of the twelve species known as causative agents, including the most important, *M. mycetomatis*, and some other rare pathogenic species (i.e., *Aspergillus* and *Curvularia*), were detected from 83% (53/64) of the soil samples. In regard to specificity and sensitivity, this study shows the advantage of multiple detections of mycetoma-causative pathogens. Our results showed that regardless of the environmental conditions, various eumycetoma pathogens are ubiquitous in the soils of these areas. In addition, the analysis showed a statistical difference between roads and drylands, which indicates that some environmental agents may affect the existence of pathogens in soil, though this requires further study. More extensive field sampling, including soil composition or climate data, may uncover the environmental preferences of mycetoma-causative fungi. Moreover, the outcome of soil fungal metabarcoding can be used to visualize a risk map of the causative agents with remote sensing data, as with previous studies using clinical data [37,38]. This can lead to robust estimates of the environmental risk factors associated with mycetoma, which have not been elucidated over its long history.

Furthermore, our findings can help with diagnosis or preventive measures for patients. Clarifying the diversity of fungi, including eumycetoma agents, in patients' villages might contribute to the accurate diagnosis of the pathogens. Our data also suggest that protective measures such as wearing shoes should be strongly promoted for people in mycetoma-endemic areas.

Based on the data obtained in this study, we constructed a system of amplicon sequencing and metabarcoding techniques targeting a group of mycetoma-causative organisms. Considering the PCR conditions, we faced a major limitation: some extracted DNA solutions were brown-colored (S2 Table), which means that the samples might have been contaminated with PCR inhibitors such as humic acids in the soil. Here, we applied a DNA extraction method developed for sedimentary soils of river bottoms [20]; however, the soil moisture contents were different for each land use, which might have influenced the PCR amplification. For subsequent MiSeq analysis targeting ribosomal ITS2, the colored DNA samples were diluted in the 1st PCR to produce accurate results. Accordingly, most target DNA was successfully amplified (i.e., reads and ASVs) compared with the same samples that were not diluted.

In this study, we focused on species-level detection of causative agents of eumycetoma. Therefore, the number of sequences that could not be identified as species might be underestimated, which means there were possibly more pathogens in the sampling sites. Thus, further identifying species from mycetoma specimens and registration in gene databases is still

needed. In our results, *Aspergillus terreus* was only identified at the species level for the ribosomal DNA ITS1 region, although the *Madurella* genus was identified in the analysis for ITS2. For pathogenic species identification, at this point, using both the ITS1 and ITS2 regions would lead to more accurate results.

## Conclusion

The metabarcoding technology-based soil sampling survey method can detect multiple causative fungi from soil samples in an endemic area, including the major pathogen, *M. mycetoma*. Applying the technology to construct a geographic distribution of causative fungi provides essential and fundamental information for preventive measures, diagnosis considering regional characteristics and fungi distribution, and the development of therapeutic agents against mycetoma.

## Supporting information

**S1 Table. Causative microorganisms of eumycetoma in the world.** We used sequences of the accession numbers below to choose universal primers.
(XLSX)

**S2 Table. Colored DNA samples after extraction.** These samples had inadequately remove soil-derived PCR inhibitors. Each letter shows a sampling village: A. Deshein, B. Sharfa Al-Mihrab, C. Al-Awia, D. Wad Hassan, E. Wad Ajeeb, F. Al-Ragal Al-Batahin, and G. Wad Al-har. NCs, negative controls at DNA extraction.
(XLSX)

**S1 Fig. Number of detected eumycetoma-causative species per land-use category.**
(EPS)

**S1 Protocol. Supplemental methods.** Supporting information of methods. Supplemental method 1. Soil sampling. Supplemental method 2. DNA extraction. Supplemental method 3. PCR amplification. Supplemental method 4. Bioinformatics.
(DOCX)

## Acknowledgments

We appreciate all the members of MRC, including but not limited to Dr. Sahar Mubarak Bakhiet, for arranging the travel and preparing all the necessary reagents and laboratory equipment at MRC. We are grateful for all the people in Sennar State who welcomed us into their villages and the community health workers who guided us around. We acknowledge Dr. Tomonori Hoshi for the ODK setting and shell and R coding assistance, Dr. Satoshi Yamamoto, and Dr. Takeshi Nabeshima for general advice on bioinformatics. We also thank Dr. Qianqian Wu, Ms. Kana Hayami, and Mr. Daiki Takeshita for supporting PCR experiments.

## Author Contributions

**Conceptualization:** Toshifumi Minamoto, Ahmed Hassan Fahal, Satoshi Kaneko.

**Data curation:** Suguru Taga, Emmanuel Edwar Siddig.

**Formal analysis:** Hiroki Hashizume, Suguru Taga, Masayuki K. Sakata.

**Funding acquisition:** Ahmed Hassan Fahal, Satoshi Kaneko.

**Investigation:** Suguru Taga, Mahmoud Hussein Mohamed Taha, Emmanuel Edwar Siddig, Ahmed Hassan Fahal, Satoshi Kaneko.

**Methodology:** Hiroki Hashizume, Suguru Taga, Masayuki K. Sakata, Toshifumi Minamoto.

**Project administration:** Ahmed Hassan Fahal, Satoshi Kaneko.

**Resources:** Ahmed Hassan Fahal.

**Software:** Hiroki Hashizume, Masayuki K. Sakata.

**Supervision:** Ahmed Hassan Fahal, Satoshi Kaneko.

**Visualization:** Hiroki Hashizume.

**Writing – original draft:** Hiroki Hashizume, Suguru Taga.

**Writing – review & editing:** Masayuki K. Sakata, Toshifumi Minamoto, Ahmed Hassan Fahal, Satoshi Kaneko.

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
