## [Decision Letter · Decision Letter 0]

24 Nov 2021

Dear Prof. Kaneko,

Thank you very much for submitting your manuscript "A novel method of an environmental study of multiple mycetoma pathogens: Metabarcoding analysis for soil DNA of the endemic area in Sudan" for consideration at PLOS Neglected Tropical Diseases. As with all papers reviewed by the journal, your manuscript was reviewed by members of the editorial board and by several independent reviewers. In light of the reviews (below this email), we would like to invite the resubmission of a significantly-revised version that takes into account the reviewers' comments. 

Dear Dr. Kaneko and colleagues:

Thanks for submitting your manuscript to PLoS Neglected Tropical Diseases. I have now received three independent reviews of your work, and as you will see, one reviewer recommended rejection, another suggested a major revision and third a minor revision. I am affording you the option of revising your manuscript according to all three reviews but understand that your resubmission may be sent to at least one new reviewer for a fresh assessment (unless the reviewer recommending rejection is willing to re-review).

There are concerns about the novelty of your methods…please explain how your approach is novel and differs from standard approaches used in the field. I realize this is one opinion and the other reviewers are less concerned about the novelty of your methodology. So in your rebuttal letter please argue for the novelty aspect of your study.

Please have an English-speaking expert help revise your manuscript. Please also work on the clarity, organization and presentation of the manuscript (several suggestions are made by the reviewers). Submit your manuscript with track-changes demonstrating that a native-speaking expert has helped with grammar and context.

Importantly, please ensure your Materials and Methods are clearly stated. The methods should be clear, concise and repeatable. Please ensure this, and make sure all relevant information and references are provided. 

I look forward to seeing your revision, and thanks again for submitting your work to PLoS Neglected Tropical Diseases.

Good luck with your revision,

-joe

We cannot make any decision about publication until we have seen the revised manuscript and your response to the reviewers' comments. Your revised manuscript is also likely to be sent to reviewers for further evaluation.

Sincerely,

Joseph James Gillespie, Ph.D.

Associate Editor

Dileepa Ediriweera

Deputy Editor

Dear Dr. Kaneko and colleagues:

Thanks for submitting your manuscript to PLoS Neglected Tropical Diseases. I have now received three independent reviews of your work, and as you will see, one reviewer recommended rejection, another suggested a major revision and third a minor revision. I am affording you the option of revising your manuscript according to all three reviews but understand that your resubmission may be sent to at least one new reviewer for a fresh assessment (unless the reviewer recommending rejection is willing to re-review).

There are concerns about the novelty of your methods…please explain how your approach is novel and differs from standard approaches used in the field. I realize this is one opinion and the other reviewers are less concerned about the novelty of your methodology. So in your rebuttal letter please argue for the novelty aspect of your study.

Please have an English-speaking expert help revise your manuscript. Please also work on the clarity, organization and presentation of the manuscript (several suggestions are made by the reviewers). Submit your manuscript with track-changes demonstrating that a native-speaking expert has helped with grammar and context.

Importantly, please ensure your Materials and Methods are clearly stated. The methods should be clear, concise and repeatable. Please ensure this, and make sure all relevant information and references are provided. 

I look forward to seeing your revision, and thanks again for submitting your work to PLoS Neglected Tropical Diseases.

Good luck with your revision,

-joe

Reviewer's Responses to Questions

**Key Review Criteria Required for Acceptance?**

**Methods**

-Are the objectives of the study clearly articulated with a clear testable hypothesis stated?

-Is the study design appropriate to address the stated objectives?

-Is the population clearly described and appropriate for the hypothesis being tested?

-Is the sample size sufficient to ensure adequate power to address the hypothesis being tested?

-Were correct statistical analysis used to support conclusions?

-Are there concerns about ethical or regulatory requirements being met?

Reviewer #1: The methods used are the regular methods used in that study and there is no new and contribution to the methods used

Reviewer #2: Firstly, a thorough review of English is necessary.

The main claims of the paper is that eumycetoma is highly frequent in Sudán and the main etiologicla agent is M. mycetomatis. These claims are very significant to meet the proposed objective.

The authors have enough fairly the literature. 

In general the manuscript is well organized, but not written clearly enough to be accessible to non-specialists.

Methods

The objective of the study is clearly established but not articulated with a clear testable hypothesis.

-The study design could be supplemented with culture to address the stated objectives.

-The population is clearly described and appropriate.

-The sample size sufficient to ensure adequate power to address the study.

-The statistical analysis used were correct to support conclusions.

-Specifically, in the sampled sites (Sennar State Suda), what are the most frequent eumycetoma agents reported? If the answer is M. mycetomatis, something is not well in the methodology. The authors tried to recover M. mycetomatis in culture? 

-Some of additional work would improve the paper. For exemple, cultivation of different soils to get eumycetoma agents and phenotipically identify these ones. This methodology is laborious but not difficult.

-The NGS is a technique that makes outstanding this paper, but unfortunately is not well explained and results no well exploited. Authors do not say if NGS was make at least by duplicate.

-Details of the methodology are no sufficient to allow the experiments to be reproduced.

-The analysis presented does not match the analysis plan. Madurella mycetomatis DNA was not detected by NGS after DNA extraction from soil.

-Results are clearly and completely presented, but are not that expected.

-The figures (Tables, Images) are of sufficient quality for clarity.

-I think authors should have done some previous experiments to standardize amplification conditions and then apply them to the NGS technique.

Reviewer #3: -Are the objectives of the study clearly articulated with a clear testable hypothesis stated?

Yes

-Is the study design appropriate to address the stated objectives?

Yes

-Is the population clearly described and appropriate for the hypothesis being tested?

Yes

-Is the sample size sufficient to ensure adequate power to address the hypothesis being tested?

Yes

-Were correct statistical analysis used to support conclusions?

Yes

-Are there concerns about ethical or regulatory requirements being met?

No

**Results**

-Does the analysis presented match the analysis plan?

-Are the results clearly and completely presented?

-Are the figures (Tables, Images) of sufficient quality for clarity?

Reviewer #1: ok

Reviewer #2: -The analysis presented does not match the analysis plan. Madurella mycetomatis DNA was not detected by NGS after DNA extraction from soil.

-Results are clearly and completely presented, but are not that expected.

-The figures (Tables, Images) are of sufficient quality for clarity.

Reviewer #3: -Does the analysis presented match the analysis plan?

Yes

-Are the results clearly and completely presented?

Yes

-Are the figures (Tables, Images) of sufficient quality for clarity?

Yes; please make sure the map employed complies with the journal copyright requirements.

**Conclusions**

-Are the conclusions supported by the data presented?

-Are the limitations of analysis clearly described?

-Do the authors discuss how these data can be helpful to advance our understanding of the topic under study?

-Is public health relevance addressed?

Reviewer #1: there is no conclusion on the study but it is general sentences not supported to the results obtained

Reviewer #2: Discussion

In Discussion, authors say that that they did not find M. mycetomatis because of some PCR inhibitors. Why was this phenomenon not present with the other fungal agents?

Conclusions

-The conclusions are not supported by the data presented, but only by the method advantage.

-The limitations of analysis considered by authors are described.

-The public health relevance is addressed.

Reviewer #3: -Are the conclusions supported by the data presented?

Yes

-Are the limitations of analysis clearly described?

Yes

-Do the authors discuss how these data can be helpful to advance our understanding of the topic under study?

Yes

-Is public health relevance addressed?

Yes

**Editorial and Data Presentation Modifications?**

Reviewer #1: (No Response)

Reviewer #2: I dont have experience about Editorial work. However, if Editorial Committee decides accept this paper I dont have modifications to recommend.

Reviewer #3: Minor Revision

**Summary and General Comments**

Reviewer #1: This paper lacks novelty. there is no new in the results of the study and it is not contribute to the readers rather than the scientific community. I ask where is the new findings or contribution in the field?. All methods are routinely used in the similar studies, also the results obtained have no impact in the scientific community

Reviewer #2: I have reviewed the manuscript entitled “A novel method of an environmental study of multiple mycetoma pathogens: Matabarcoding análisis for soil DNA of the endemic área in Sudan”.

 The authors use a relatively novel technique and this is remarkable and valuable. However, the wording and clarity must be improved especially in the methodology. Unfortunately authors did not recover M. mycetomatis DNA, as the main eumycetoma agent in Sudan. If the authors improve the writing and clarity of the English (the discussion is fine), the document is salvageable.

Reviewer #3: The authors performed an analysis of soil samples in several locations in Sudan with the objective of detecting eumycetoma causing agents. The study is well performed and the methodology thoroughly detailed. The findings are interesting and relevant due to the scarcity of environmental studies of these fungi. Please include in the methodology section the month and usual weather when the soil samples were retrieved as these may influence future studies that aim to isolate or detect these microorganisms. Please include in the discussion section the reasons you believe of M. mycetomatis was not found.

PLOS authors have the option to publish the peer review history of their article (what does this mean?). If published, this will include your full peer review and any attached files.

Reviewer #1: Yes: Tarek A. A. Moussa

Reviewer #2: No

Reviewer #3: Yes: JA Cardenas-de la Garza
---

## [Decision Letter · Decision Letter 1]

23 Feb 2022

Dear Prof. Kaneko,

We are pleased to inform you that your manuscript 'Detection of multiple mycetoma pathogens using fungal metabarcoding analysis of soil DNA in an endemic area of Sudan' has been provisionally accepted for publication in PLOS Neglected Tropical Diseases.

Best regards,

Joseph James Gillespie, Ph.D.

Associate Editor

Dileepa Ediriweera

Deputy Editor

Dear Dr. Kaneko and colleagues:

Thanks for revising your manuscript based on the concerns raised by the reviewers. I now believe that your manuscript is suitable for publication. Congratulations! I look forward to seeing this work in print, and I anticipate it being an important resource for groups studying Mycetoma pathogens as well as approaches using metabarcoding fungi. Thanks again for choosing PLoS Neglected Tropical Diseases to publish such important work.

Best,

-joe

Reviewer's Responses to Questions

**Key Review Criteria Required for Acceptance?**

**Methods**

-Are the objectives of the study clearly articulated with a clear testable hypothesis stated?

-Is the study design appropriate to address the stated objectives?

-Is the population clearly described and appropriate for the hypothesis being tested?

-Is the sample size sufficient to ensure adequate power to address the hypothesis being tested?

-Were correct statistical analysis used to support conclusions?

-Are there concerns about ethical or regulatory requirements being met?

Reviewer #2: Fort he second time I have reviewed the manuscript entitled “Detection of multiple mycetoma pathogens using fungal metabarcoding analysis of soil DNA in an endemic area of Sudan.” The document looks clearer and better written.

Methods

The objective of this study is clear, as well as the study design. The geographical sites to study are well described. The authors do not propose a hypothesis. Considering the authors use an innovative but very expensive technology, the sample size is adequate, as well as the statistical analyse.

Reviewer #3: (No Response)

**Results**

-Does the analysis presented match the analysis plan?

-Are the results clearly and completely presented?

-Are the figures (Tables, Images) of sufficient quality for clarity?

Reviewer #2: The presented analysis match well with analysis plan. Results are clearly and completely presented, both in text and figures. Figures are of sufficient quality and clarity.

Reviewer #3: (No Response)

**Conclusions**

-Are the conclusions supported by the data presented?

-Are the limitations of analysis clearly described?

-Do the authors discuss how these data can be helpful to advance our understanding of the topic under study?

-Is public health relevance addressed?

Reviewer #2: Conclussions are supported by the data presented. Limitations of analysis are not mentioned by authors. The authors discuss how the knowledge of the distribution of eumycetoma agents can contribute to understanding the population's risk of developing the disease. These data can be helpful.

The relevance in public health is highlighted, indicating that Sudan is a country with a high rate of eumycetoma.

Reviewer #3: (No Response)

**Editorial and Data Presentation Modifications?**

Reviewer #2: I suggest ACCEPT the manuscript for publication, with a minor change: In Conclusión, line 334, Did the authors mean M. mycetomatis?

Reviewer #3: Accept

**Summary and General Comments**

Reviewer #2: After a second review, the manuscript is clearer and better written. Authors performed a novel technique with results that reinforce the information available on the etiological agents of mycetoma.

Reviewer #3: Great job.

PLOS authors have the option to publish the peer review history of their article (what does this mean?). If published, this will include your full peer review and any attached files.

Reviewer #2: No

Reviewer #3: **Yes: **JA Cardenas-de la Garza

---

## [Editor Report · Acceptance letter]

7 Mar 2022

Dear Prof. Kaneko,

We are delighted to inform you that your manuscript, "Detection of multiple mycetoma pathogens using fungal metabarcoding analysis of soil DNA in an endemic area of Sudan," has been formally accepted for publication in PLOS Neglected Tropical Diseases.

Best regards,

Shaden Kamhawi

co-Editor-in-Chief

Paul Brindley

co-Editor-in-Chief
